**Perspective**

# Redefining the photic zone: beyond the autotroph-centric view of light in the ocean
Thomas W. Davies [1] ✉ & Tim Smyth [2,3]

Traditional measures of the photic zone have remained focused on autotrophs, limiting understanding of how changing marine lightscapes impact heterotrophs that use light as a resource or an environmental cue. We propose a new photic zone definition that encompasses all biological processes influenced by celestial light, and a new measure of photic zone depth, the minimum light intensity that elicits biological responses. This approach allows photic zone measures to be inclusive of all marine photobiology driven by sunlight, moonlight, or starlight, and enables urgently needed exploration of the nature, extent and ecological implications of changing marine lightscapes.

The photic zones of the oceans support 90% of all marine life and provide critical ecosystem services on a global scale, including sustaining fisheries[1,2] and regulating nutrient and carbon cycles[3,4]. Ocean lightscapes, and the photic zones they contain, are, however undergoing significant changes[5–8]. Widespread greening[7] and darkening[8–12] of surface waters are being driven by increasing precipitation, deglaciation, reducing sea ice, land-use change, and shifts in global ocean circulation, among other factors. These changes have led to reductions in photic zones by more than 50 m across 9% of the global ocean in the early 21st century[8]. Simultaneously, coastal urbanisation[13] and the recent widespread adoption of white Light Emitting Diodes[14] has increased artificial light emissions into the sea at night[15–17], brightening nightime underwater habitats and impacting a broad array of species and ecosystems[6].

Interest in changing marine lightscapes and their impact on species, ecosystems and the services they provide to humans is rapidly growing. This interdiciplinary field increasingly brings together sensory biologists, remote sensing scientists and optical oceanographers to investigate how photic zones are changing and the biological implications of these shifts. This evolving research landscape necessitates a redefinition of the photic zone concept. Originally concieved in relation to primary productivity, we argue that the conventional definitions, measurements and conceptualisations of the photic zone hinder scientific progress in understanding changing marine lightscapes. We propose a broader definition that encompasses all photobiological responses to celestial light cues, facilitating a more comprehensive exploration of temporal and spatial changes in photic zones in relation to both optical water properties and surface light conditions. The widely accepted metric for defining photic zone depth—the 1% light level—is critically evaluated alongside alternative measures based on a suite of desirable properties. Finally, we introduce a new framework for categorizing the ocean's major light zones and recommend thresholds for their measurement.

## Review

**The photic zone definition.** Currently, the scientific community defines three distinct zones of the ocean in relation to the penetration of light. The uppermost 'euphotic zone' is conventionally considered the region where sufficient light penetrates to enable photosynthesis to occur. It is often referred to as around 200 m depth on average, although in reality this depth varies daily and seasonally across latitudes with changes in the incident sea surface irradiance and variability in the optical properties of the water column (chlorophyll-*a*, suspended particulate matter, coloured dissolved organic matter, etc.). The terms 'photic zone' and 'euphotic zone' are commonly conflated and used interchangeably. Below the euphotic zone is the dysphotic or twilight zone, where light is present but not in sufficient quantities for photosynthesis to exceed respiration. The deepest layer is termed the aphotic zone where no appreciable light penetrates.

Within the marine environment, the near ubiquitous process of photosynthesis by phytoplankton, has resulted in the photic zone being defined using autotroph-centric constructs[18,19]. Primary production utilises the wavelengths of light between 400 and 700 nm in the electromagnetic range known as Photosynthetically Active Radiation (PAR) or alternatively as Photosynthetically Usable Radiation (PUR)[20] typically measured in μmol photons m$^{-2}$ s$^{-1}$ (1 μmol photons m$^{-2}$ s$^{-1}$ is equivalent to 0.217 W m$^{-2}$ daylight). The global distribution of photosynthesis results in around 50 Gt C yr$^{-1}$ being drawn from the atmosphere[21] with approximately 10 Gt C yr$^{-1}$ being exported to the deep ocean via the biological carbon pump[22]. It is the ubiquity and magnitude of this marine process, coupled

[1]School of Biological and Marine Sciences, University of Plymouth, Plymouth, UK. [2]Plymouth Marine Laboratory, Prospect Place, Plymouth, UK. [3]Centre for Geography and Environmental Science, Department of Earth and Environmental Sciences, University of Exeter, Penryn Campus, Penryn, Cornwall, UK. ✉e-mail: thomas.w.davies@plymouth.ac.uk

with it being one of the most studied and understood, that has likely led to the autotroph-centric view of photic zones.

**Limitations of use**. Photosynthesis is currently the critical defining feature of the photic zone[18,19,23,24]. Photosynthesis is not, however, implied by the etymology of any term used to describe a zone of the oceans in relation to light. The word 'photic' derives from the Greek 'phos' meaning 'light', 'eu' derives from the Greek 'good' commonly used scientifically as 'true', 'dys' means 'bad' or 'disordered', and α (a) means 'without'. This presents an important question, 'how do we define the true lit ocean?'.

Photosynthesis is but one biological process that relies on natural light entering the surface ocean. It is very insensitive to light compared to myriads of other processes that make use of sunlight and moonlight (Table 1)[25–28]. These processes are critical to the life histories of marine organisms and their population demography. Unlike photosynthesis which uses light as energy, they are guided by the information contained in light signals[29,30]. This information is detected by specialised light sensitive organs or regions including eyes, ocelli, pits and dense regions of light sensitive cells[27,31–33], and even pigments that regulate photosynthesis in autotrophs[34]. Importantly, adaptations that enable organisms to 'see' light tend to be far more sensitive to light than photosynthesis because they quantify radiance (the light received per unit angle) rather than the more diffused property irradiance (light striking a surface per unit area). Numerous examples of this sensitivity exist including the diel vertical migration of zooplankton[35]—'the largest daily migration of biomass on the planet'[36,37] that response to moonlight during the Arctic winter[36], synchronised broadcast spawning in relation moonlight cycles[26,28,38–42], and settlement site selection in numerous marine invertebrate larvae[25,43–47].

By focusing only on photosynthesis, the currently held definition of the photic zone ignores these equally important and phylogenetically far more widespread adaptations to light[48]. It has also inadvertently created a very 'daylight centric' view of photobiology in the oceans when many marine species (especially invertebrates) are actually nocturnal. Properties such as $K_d(490)$—a measure of the diffuse attenuation of light (at 490 nm)—are rarely used to explore how changing light fields impact photobiological processes other than photosynthesis, and even rarer still are they used to explore changing light fields in the oceans at night. The definition is also unsuitable for use outside of ocean colour remote sensing and the conventional questions explored by optical oceanography. How does a sensory biologist define the photic zone when its current definition is restricted to photosynthesis?

**A new photic zone definition**. We propose a broader definition that distinguishes the photic zone from the euphotic zone as '*the depth to which light from celestial light sources (sunlight, moonlight or starlight) is sufficient for photobiological processes to occur in response to these light sources*'. This definition is inclusive of all biological processes utilizing natural light incident at the sea surface during any time of day or year, and in any location. It is also inclusive of the euphotic (epipelagic) and dysphotic (mesopelagic) zones, avoiding the need to redefine them while providing clarity in the distinction between the terms 'photic' and 'euphotic', which are often conflated. The specification of the light sources also distinguishes the photic zone from biological sources of illumination that occur in the aphotic and dysphotic zones[31].

Our intention is that adopting such a definition will help steer the scientific community towards a broader recognition of the importance of light in the oceans that is less orientated around photosynthesis. It recognises the importance of nocturnality in marine ecosystems and moonlight cycles in marine organisms life histories. Adopting such a broad definition however, poses another equally important question, 'How should we quantify the photic zone?'.

**Current convention for quantifying Z$_{photic}$**. Given the autotroph-centric view towards defining the photic zone, it is perhaps unsurprising that the widely adopted approach for quantifying the photic zone depth

($Z_{photic}$) is based on photosynthesis. This quantity - commonly referred to as the 1% threshold - is the depth to which the Photosynthetically Active Radiation (PAR) irradiance falls to 1% of its value at the sea surface. It is commonly held that the 1% equates roughly to the irradiance at which appreciable photosynthesis can occur (Table 2). Note that while the 1% threshold measure is often used to define the euphotic zone, the widespread conflation of the terms euphotic and photic has led to the perception that it is also the 'standard' measure of photic zone depth.

Aside from being centred around autotrophic processes, the 1% threshold is a poor quantity even for defining the depth to which primary production occurs. The arbitrary nature of the 1% threshold dislocates the measure from its absolute value which in reality is highly variable in space and time. Being a relative quantity that is unimpacted by surface light irradiance, the 1% threshold is not actually a measure of light availability, but of light attenuation by the optical constituents of the water column. If the light intensity over a body of water with fixed optical properties increased by any amount, the 1% threshold depth remains the same, regardless of changes in light irradiance at that depth (Fig. 1). Using this definition the depth of the photic zone in a body of water with fixed optical properties is the same regardless of the time of day or year, whether illuminated by sunlight, moonlight or starlight (Fig. 1).

The widespread adoption of the 1% threshold may seem counterproductive since it prohibits exploration of how the motions of the sun and moon impact the photic zones and biological processes within them, how photic zones change through the months and seasons with the waxing and waning of light, and how all these processes depend on an organisms latitudinal position. The relative nature of the measure also means that it has no biological relevance. 1% of surface irradiance is not a biological quantity, and so the measure cannot be used to explore either how photic zones are changing or how these changes impact on the biology of the oceans. As a result, the 1% light level is largely biologically irrelevant and has the disadvantage of defining photic zone depths with varying irradiance levels across space and time, as its depth remains constant regardless of surface irradiance. Given this limitation, a biologist might reasonably question how the 1% light level became the standard measure of photic zone depth.

**A brief history of the current convention**. In reference to quantifying euphotic zone depth, Kirk[19] states that:

"*A useful, if approximate, rule-of-thumb in aquatic biology is that significant phytoplankton photosynthesis takes place only down to that depth, $Z_{eu}$, at which the downwelling irradiance of PAR falls to 1% of that just below the surface. That layer within which $E_d(PAR)$ falls to 1% of the subsurface value is known as the euphotic zone.*"

Within this statement, there are five qualitative and unsubstantiated phrases: "*… useful, … approximate, rule-of-thumb … significant … just below …*"

Unfortunately, the Kirk[19] textbook definition has arbitrarily led to the almost ubiquitous conflation of the euphotic zone (and the photic zone by association) with the 1% light level. But where did the 1% threshold originate from?

Ryther[18] in his seminal paper on Primary Productivity based on seasonal measurements at Woods Hole states the following in a footnote when referring to the 'entire euphotic zone':

"*Defined here as limited by the depth of penetration of 1% of the radiation incident to the surface. This is a constant depth (in water of the same transparency) but receives illumination which varies with the surface radiation. Thus the euphotic zone, as used here, has no biological significance other than defining the water mass below which no appreciable photosynthesis can occur.*"

Striking in this statement is the admission that the 1% light level has no biological significance beyond defining the depth of appreciable photosynthesis. Yet the relative nature of the quantity also means it has no direct relevance to photosynthesis. Further, Ryther[18] assumes that the ratio of Photosynthesis to Respiration (P:R) is 10:1. Yet the paper clearly shows that in the winter months, with short days and low altitude sun, that respiration

**Table 1 | Threshold light sensitivities for marine species' behavioural responses**

| Group | Taxon | Life stage | Response | Threshold Light Sensitivity (µW m⁻²) | Light source | Reference | Z (m) Solar | Lunar |
|---|---|---|---|---|---|---|---|---|
| Arthropoda | *Calanus* spp. | Adult (f) | Phototaxis (−ve), DVM | $1.02 \times 10^{-1}$ | LED (white) | Bâtnes et al. (2015)[50] | 200 | 40 |
| Arthropoda | *Calanus* spp. | Adult (m) | Phototaxis (−ve), DVM | 1.02 | LED (white) | Bâtnes et al. (2015)[50] | 200 | 10 |
| Arthropoda | *Semibalanus balanoides* | Larvae | Phototaxis (varied) | $1.01 \times 10^{-1}$ | 2150 K | Crisp and Ritz (1973)[43] | 200 | 40 |
| Arthropoda | *Pleuromamma xiphias* | Adult | Phototaxis (−ve), DVM | $2.61 \times 10^{-1}$ | 100 W quartz halogen | Buskey et al. (1989)[52] | 200 | 30 |
| Arthropoda | *Uca* spp. | Megalope | Phototaxis (−ve) | 3.62 | 1000 W incandescent lamp | Tankersley et al. (1995)[53] | 150 | 1.5 |
| Arthropoda | *Meganyctiphanes norvegica* | Adult | Abdominal flexion | 3.62 | White slide projector | Myslinski et al. (2005)[54] | 150 | 1.5 |
| Arthropoda | *Pasiphaea multidentata* | Adult | Abdominal flexion | 3.62 | White – Slide projector | Myslinski et al. (2005)[54] | 150 | 1.5 |
| Arthropoda | *Eusergestes arcticus* | Adult | Abdominal flexion | $3.62 \times 10^{1}$ | White slide projector | Myslinski et al. (2005)[54] | 150 | 0 |
| Arthropoda | *Pleuromamma gracilis* | Adult | Phototaxis (−ve), DVM | $5.07 \times 10^{1}$ | 100 W quartz halogen | Buskey et al. (1989)[52] | 100 | 0 |
| Chordata | *Ascidia mentula* | Larvae | Phototaxis (−ve) | $2.17 \times 10^{2}$ | 20 W fluorescent tubes | Svane and Dolmer (1995)[55] | 100 | 0 |
| Arthropoda | *Callinectes sapidus* | Larvae | Phototaxis (+ve) | $3.62 \times 10^{2}$ | 750 W incandescent lamp | Forward Jr et al. (1995)[56] | 100 | 0 |
| Arthropoda | *Pontella karachiensis* | Adult | Phototaxis (−ve) | $6.78 \times 10^{2}$ | fluorescent lamp | Manor et al. (2009)[57] | 100 | 0 |
| Annelida | *Phragmatopoma caudata* | Larvae | Phototaxis (+ve) | $6.88 \times 10^{2}$ | 300 W incandescent lamp | McCarthy et al. (2002)[58] | 100 | 0 |
| Chordata | *Hippoglossus hippoglossus* | Larvae | Phototaxis (+ve) | $2.01 \times 10^{3}$ | fluorescent tube | Naas and Mangor-Jensen (1990)[59] | 90 | 0 |
| Chordata | *Ascidia callosa* | Larvae | Change in swimming | $2.17 \times 10^{3}$ | 100 W incandescent bulb | Young and Chia (1985)[60] | 90 | 0 |
| General Phytoplankton | | | Photosynthesis light minima | $2.17 \times 10^{3}$ | Sunlight | Raven et al. (2000)[61] | 90 | 0 |
| General Phytoplankton | Arctic Ocean | | Compensation irradiance | $8.70 \times 10^{3}$ | Sunlight | Hoppe et al. (2024)[62] | 70 | 0 |
| Platyhelminthes | *Prosthecereaus crozieri* | Larvae | Phototaxis (+ve) | $6.67 \times 10^{4}$ | 300 W incandescent | Johnson and Forward Jr (2003)[63] | 45 | 0 |
| Chordata | *Cyanea capillata* | Larvae | Phototaxis (−ve) | $7.25 \times 10^{4}$ | 20 W fluorescent tubes | Svane and Dolmer (1995)[55] | 45 | 0 |
| Chordata | *Polyandrocarpa zorritensis* | Larvae | Phototaxis (+ve) | $1.30 \times 10^{5}$ | Sunlight | Forward et al. (2000)[64] | 40 | 0 |
| Porifera | *Amphimedon queenslandica* | Larvae | Phototaxis (−ve) | $2.17 \times 10^{5}$ | 3200 K Fibre Optic | Leys and Degnan (2001)[65] | 30 | 0 |
| Heterokontophyta | *Phaeodactylum tricornutum* | | Compensation irradiance | $2.17 \times 10^{5}$ | Fluorescent lamp | Geider et al. (1986)[66] | 30 | 0 |
| Heterokontophyta | Skeletonema costatum | | Compensation irradiance | $2.39 \times 10^{5}$ | Cool-white fluorescent | Langdon (1987)[67] | 30 | 0 |
| Annelida | *Spirobranchus giganteus* | Larvae | Phototaxis (+ve) | $3.62 \times 10^{5}$ | 5600 K | Marsden (1986)[68] | 25 | 0 |
| General Phytoplankton | North Atlantic | | Compensation irradiance | $5.03 \times 10^{5}$ | Sunlight | Marra (2004)[69] | 20 | 0 |
| Chordata | *Polyandrocarpa zorritensis* | Larvae | Phototaxis (+ve) | $1.05 \times 10^{6}$ | Sunlight | Forward et al. (2000)[64] | 10 | 0 |
| Heterokontophyta | *Olisthodiscus luteus* | | Compensation irradiance | $1.96 \times 10^{6}$ | Cool-white fluorescent | Langdon (1987)[67] | 4.5 | 0 |
| Annelida | *Serpula vermicularis* | Larvae | Phototaxis (−ve) | $2.17 \times 10^{6}$ | 'White' incandescent | Young and Chia (1982)[70] | 3.5 | 0 |
| General Phytoplankton | Southern Ocean | | Compensation irradiance | $2.77 \times 10^{6}$ | Sunlight | Regaudie-de-Gioux and Duarte (2010)[71] | 2 | 0 |
| Chordata | *Pyura chilensis* | Larvae | Phototaxis (−ve) | $3.62 \times 10^{6}$ | Sunlight | Manríquez and Castilla (2007)[72] | 0.5 | 0 |
| Annelida | *Galeolaria caespitosa* | Larvae | Phototaxis (−ve) | $3.62 \times 10^{6}$ | 5600 K | Marsden (1988)[73] | 0.5 | 0 |

Values are given for taxa whose minimum light sensitivities have been quantified using broad spectrum light sources enabling reliable conversion from photons to irradiance using sunlight as a proxy spectrum. Values of the depth (Z) at which each response can occur are given for sunlight and moonlight propagating through hypothetical case water scenario for illustration in Fig. 1. See Fig. 1 legend for details.

**Table 2 | Properties of alternative photic zone measures**

| $Z_{photic}$ measure | 1% threshold | Compensation depth | Photosynthesis minima | $Z_{photic}$(max) |
|---|---|---|---|---|
| Definition | 1% of the surface value | Light required for photosynthesis to equal respiration | Minimum light required for photosynthesis | Minimum light known to elicit a photobiological response to a celestial light source. |
| Measure | 1% of surface irradiance | $0.2\ W\,m^{-2}$ [a] | $2174\ \mu W\,m^{-2}$ daylight [b] | $0.027\ \mu W\,m^{-2}$ @ 490 nm [c] or broadband ($0.1\ \mu W\,m^{-2}$) [d] isolume |
| Absolute quantity | No | Yes | Yes | Yes |
| Variable with surface irradiance | No | Yes | Yes | Yes |
| Biologically meaningful | No | Yes | Yes | Yes |
| Inclusive of all autotrophs | No | Yes | Yes | Yes |
| Inclusive of all light sensitive spp. | No | No | No | Yes |

[a]Taken as the minimum daylight irradiance required for photosynthesis to exceed respiration (Table 1), excluding sub-polar ice species.
[b]Taken from Raven et al. (2000)[61].
[c]Taken as the average quantity of 488 nm and 525 nm unidirectional light required to elicit a phototactic response in adult female Calanus copepods at 488 nm and 525 nm (Båtnes et al., 2013)[50].
[d]Taken as the quantity of white light required to elicit diel vertical migration reported by adult female calanus copepods (Båtnes et al., 2013)[50].

outstrips productivity within the euphotic zone when quantified using the 1% light level. This confirms that even for primary production, the 1% light level was understood to lack biological meaning at the time of its conception.

The continued use of the 1% light level to define photic zone depth appears not to be based on any convincing scientific argument for why it is a useful or desirable property to measure, but instead based what has become standard practice, convention or tradition. So, what should be the properties of an ideal measure of photic zone depth?

**An ideal photic zone measure.** We identify five desirable properties for a photic zone measure to be scientifically insightful (Table 2). Whether defined using photosynthesis or photobiological processes more broadly as we do here, it is clear that biology is a central tenet of the photic zone concept. This is justified by the magnitude of the biological processes it is quantified in relation to (for example, primary productivity or diel vertical migration) and their contributions to global carbon and nutrient cycles. The oceans are also clearly divided into zones based on the distribution of marine organism adaptions to light both at coarse but also at finer spatial scales (for example, the vertical structuring of coral reef assemblages with light intensity and spectra[44]). Given how integral biology is for both the purpose and use of the photic zone concept, biological meaning should be integral to its definition. Quantifying the photic zone in a biologically meaningful way however, presents two important challenges that were conveniently overcome by the 1% light level but are nonetheless critical for the usefulness of the photic zone definition.

First, to be biologically meaningful the quantify must be measured in absolute as opposed to relative terms[49]. As evidenced with the 1% light level, a relative quantity has no intrinsic biological relevance because it cannot be equated to a fixed quantity of light at which any particular biological process occurs. Absolute measures vary with surface light intensity meaning that the depth of the photic zone can also vary in space and time. While this is arguably a desirable property for exploring how photic zones change, the spatiotemporal variability of the photic zone makes its depth a more dynamic property. The notional 200 m average depth of the euphotic zone is a poorer description for photic zones quantified in absolute terms compared to those quantified using the 1% light level and would need to be revisited.

Secondly, if photic zones are to be measured in biological terms, then what aspect of biology should be used? Even for autotrophs, alternatives to the 1% light level exist. The compensation depth—the depth where the amount of oxygen produced during photosynthesis matches that consumed by respiration—is quantifiable in absolute terms (Table 2). Compensation irradiances have been quantified for numerous species of marine algae (Table 1). Similarly, a commonly accepted threshold irradiance for photosynthesis may also serve as a useful indicator of the depth to which light is biologically useful to autotrophs (Table 2).

Both sets of values are biologically meaningful absolute quantities that allow for photic zone depth to vary both with surface irradiance and the optical properties of the water column. They are, however, autotroph-centric measures of photic zone depth that fail to recognise the importance of other light sensitive biological processes and species, and the nocturnal biology of marine ecosystems. With such a wide array of photobiological adaptations to choose from however, no one species' response to light will be reflective of all species' responses. One approach might be to measure isolumes that are specific to the species or processes being investigated[48]. Such an approach would be most suitable in specific cases and so the measure of photic zone depth becomes a flexible rather than a fixed concept.

In many situations, however, investigators might seek a 'catch all' measure of photic zone depth that can be used to investigate general changes in the property at large spatial and temporal scales. In this instance the measure would need to be representative or inclusive (the two are juxtaposed).

Examples of representative measures include a mean, median or other percentile absolute light quantity that elicits biological responses in marine species. Such a measure of general photic zone depth would ideally be

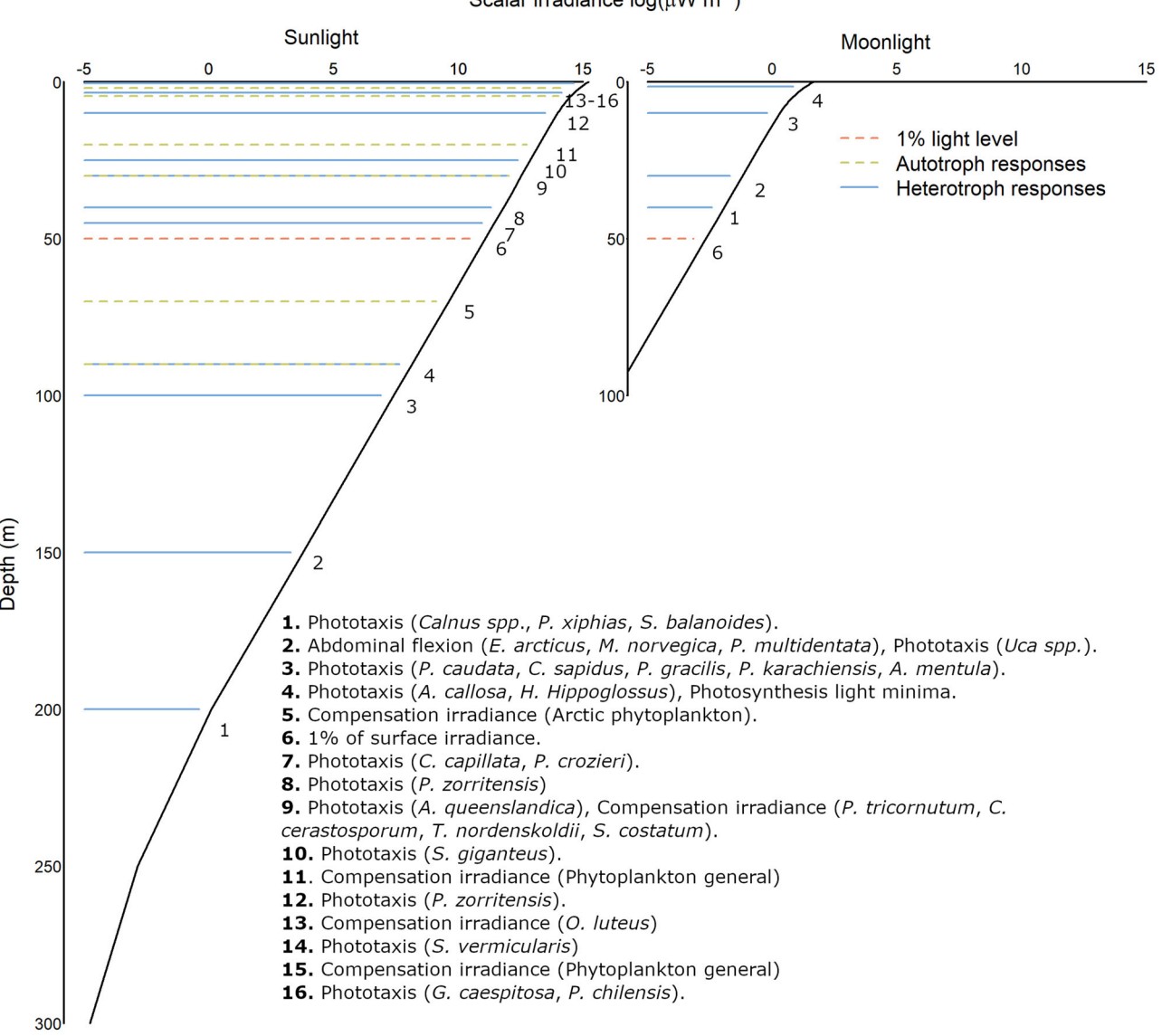

**Fig. 1 | A comparison of the depths at which marine taxa can respond to moonlight and sunlight in a hypothetical case water scenario.** Surface irradiances and spectral distributions were derived for midday on 24/6/2014 (Sunlight) and full moon on 14/04/2014 (Moonlight) for 50°N, 4°W using the approach of (Smyth et al., 2022). Spectral irradiances were propagated through a hypothetical water column of 0.3 mg m⁻³ *chl*-a with a surface wind speed of 5 ms⁻¹, using the HYDROLIGHT (Mobley, 1995) model, and integrated to provide total irradiance at 0.5 m depth bins from $0-10$ m, 5 m depth bins from 10 to 50 m, 10 m depth bins from 50 to 100 m and 50 m depth bins from 100 to 300 m. Depths ($Z$) for sunlight and moonlight are given in Table 1. The natural logarithm of scalar irradiance is presented so that declines over a greater depth range are visible.

quantified from multiple species responses to light and representative of multiple photobiological processes. The number of marine taxa for which light sensitivities have been quantified is, however, extremely limited (Table 1) and unlikely to yield a representative measure. This lack of basic biological understanding requires considerable attention from the scientific community. Such a measure would also require revising as new species light sensitivities were quantified, and poses difficult questions about imbalances in the representation of particular phyla, processes or habitats in its definition (Table 1). In practice, a representative light threshold that elicits a biological response risks introducing a less conspicuous but nonetheless equally problematic form of bias into our understanding of photic zones as those that focus exclusively on photosynthesis.

The alternative to a representative measure is one that is inclusive of the majority, if not all taxa[48]. The minimum amount of light that elicits a biological response in a species is inclusive of all species whose light sensitivities have been quantified. Under such a definition all species whose light sensitivities have been quantified would occupy the photic zone, but the depth of the photic zone is the maximum that light can elicit a biological response in any. This measure has a number of desirable properties (Table 2). It is biologically meaningful, an absolute quantity and inclusive of different species and photobiological processes. It is also the measure that most closely aligns with our proposed definition of the photic zone "*the depth to which light from celestial light sources (sunlight, moonlight or starlight) is sufficient for photobiological processes to occur in response to these light sources*".

**An alternative photic zone measure.** The 1% threshold does not meet any of the five desirable properties of a photic zone measure that we identify (Table 2). While the compensation depth and the minimum light required for photosynthesis are both desirable for measuring photic zones in relation to autotrophs, it is clear that these measures ignore the high light sensitivity of many heterotrophic species (Fig. 1, Table 1). An alternative measure has previously been used to define the depth of biologically important artificial light at night using the photosensitivity

minima of Calanus copepods[50] to broadband red, green and blue light[15]. *Calanus* species perform daily vertical migrations driven by sunlight and moonlight[51]. During dark periods, they ascend to surface waters to feed on phytoplankton, while during light periods, they descend to deeper waters to evade predators, consistently occupying a specific photic niche[50]. While other zooplankton species may contribute more to this Diel Vertical Migration in extent and biomass, the sensitivity of *Calanus* to celestial light sources coupled with the high level of detail with which this sensitivity has been quantified make it ideal for measuring the depth of the photic zone. Due to their extreme light sensitivity, *Calanus* registers comparatively deep absolute photic zone depths relative to many species, making this an inclusive rather than a representative measure of photic zone depth. Using a highly sensitive species like *Calanus* aligns with the precautionary principle, as most taxa are likely to be less photosensitive (Table 1, Fig. 1). This does not however mean that all taxa will be less sensitive to celestial light sources than *Calanus*. As our understanding of the sensitivity of marine taxa to light from the surface ocean deepens, more sensitive species may be discovered that bring the inclusivity of the *Calanus* threshold into question. The photosensitivity of *Calanus* may also be refined with alternative experimental approaches.

Davies and Smyth[8] determined the maximum photic zone depth [$Z_{photic}$(max)] as the point where irradiance at 490 nm matches the minimum light intensity at 490 nm required to trigger diel vertical migration in *Calanus* spp.[50]. The green and blue light sensitivities of Calanus reported[50] are particularly useful when calculating photic zone depths using remote or in situ collected $K_d$(490) data[15] as the median value of the $\lambda_{max}$ reported for the blue (455 nm) and green (525 nm) light sources used is 490 nm. Based on the light sensitivities of adult female Calanus copepods[50] this gives a threshold sensitivity to 490 nm light as $0.11 \times 10^{-6}$ µmol photons m$^{-2}$ s$^{-1}$ or 0.027 µW m$^{-2}$. When modelling light propagation hyperspectrally and reintegrating to provide a total irradiance, sensitivities to broad spectrum white light or natural light sources may be more appropriate to measure the depth of the photic zone (Table 1). $Z_{photic}$(max) in this context would be equivalent to the depth where light intensity was equal to or exceeded $0.47 \times 10^{-6}$ µmol photons m$^{-2}$ s$^{-1}$ or 0.1 µW m$^{-2}$ white light, a threshold used previously in studies of the extent of biologically important artificial light at night[16]. We propose these two measures of $Z_{photic}$(max) as the most informative 'catch all' measures of photic zone depth. They are measured in absolute terms, biologically relevant and inclusive of the majority of photobiological responses measured including all associated with photosynthesis and primary production in autotrophic species.

## Outlook

In this Perspectives article, we have provided a detailed critique of the current definition and measures of the photic zone and suggested a new definition and a measure that satisfy a number of desirable properties. Here, we integrate this new definition into the existing framework for describing the major light zones of our oceans as follows:-

**Photic zone**. Definition: '*The depth to which light from celestial light sources (sunlight, moonlight or starlight) is sufficient for photobiological processes to occur in response to these light sources. The photic zone comprises two sub-zones, the euphotic zone and the dysphotic zone.*'

Measure: The depth above which light from celestial sources is sufficient (0.1 µW m$^{-2}$) to stimulate measurable behavioural responses from *Calanus* spp. copepods.

**Euphotic zone**. Definition: '*The depth to which light is sufficient for the rate of photosynthesis to exceed the rate of respiration.*'

Measure: The compensation depth for phytoplankton where irradiances from celestial light sources are equal to or exceed 0.2 W m$^{-2}$.

**Dysphotic zone**. Definition: *The depths where light is insufficient for the rate of photosynthesis to exceed the rate of respiration, but where other photobiological processes responsive to celestial light sources (sunlight, moonlight or starlight) can still occur.*'

Measure: Depths where irradiances from celestial light sources fall below 0.2 W m$^{-2}$ but remain equal to or in excess of 0.1 µW m$^{-2}$.

**Aphotic zone**. Definition: "*The depths where light from celestial light sources (sunlight, moonlight or starlight) is not sufficient for photobiological processes to occur in response to these light sources*".

Measure: The depth where irradiances from celestial light sources fall below 0.1 µW m$^{-2}$ s$^{-1}$.

Marine lightscapes are undergoing significant change. The open ocean is becoming darker due to increased attenuation of sunlight, moonlight, and starlight in surface waters, while coastal and offshore developments are brightening nighttime marine environments through rising artificial light emissions, and sea ice decline is brightening the polar oceans. Although there is an urgent need to understand how these shifts impact the photobiology of marine species, progress is increasingly constrained by the traditional concept and measurement of the photic zone. Here, we propose an alternative definition and measurement framework that encompasses all photobiological processes. By adopting this revised approach, researchers can more effectively investigate the fundamental drivers of diel light niches in marine ecosystems[48], assess their transformations in the 21st century, and evaluate their ecological consequences and impacts on the essential ecosystem services provided by photic zones.

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

## Acknowledgements

This work was supported by the Natural Environment Research Council (grant numbers NE/S003533/2 and NE/S003568/1 awarded to T.W.D. and T.S.).

## Author contributions

T.W.D. and T.S. conceptualised the paper and co-authored its contents, including the analysis and presentation of Tables and Figures.

## Competing interests

The authors declare no competing interests.
