## [Transparent Peer Review file · Communications Earth & Environment]

Redefining the Photic Zone: beyond the autotroph-centric view of light in the ocean

Corresponding Author: Dr Thomas Davies

Version 0:

Decision Letter:

Dear Dr Davies,

Your manuscript titled "Redefining the Photic Zone" has now been seen by our reviewers, whose comments appear below. In light of their advice we are delighted to say that we are happy, in principle, to publish a suitably revised version in Communications Earth & Environment.

We therefore invite you to revise your paper one last time to address the remaining concerns of our reviewers. At the same time we ask that you edit your manuscript to comply with our format requirements and to maximise the accessibility and therefore the impact of your work.

EDITORIAL REQUESTS:

****Please take care to match our formatting and policy requirements. We will check revised manuscript and return manuscripts that do not comply. Such requests will lead to delays. ****

SUBMISSION INFORMATION:

OPEN ACCESS:

Communications Earth & Environment is a fully open access journal. Articles are made freely accessible on publication. For further information about article processing charges, open access funding, and advice and support from Nature Research, please visit <https://www.nature.com/commsenv/open-access>

Link Redacted

Best regards,

Alice Drinkwater, PhD
Associate Editor
Communications Earth & Environment
Consulting Editor
Communications Sustainability

REVIEWERS' COMMENTS:

Reviewer #1 (Remarks to the Author):

In "Redefining the Photic Zone" Davies and Smyth insightfully review the concept of the photic zone. First, they argue that historical and coincidental biases have led to a murky and biologically irrelevant definition of this ubiquitous term, based on a "1% of sea surface irradiance" rule of thumb. Second, they make the case that photobiology is not limited to photosynthesis and that other processes of enormous ecological importance (e.g., vertical migration of zooplankton) also use light. However, the current photosynthesis-centric view of the photic zone is problematic for how we define, study and discuss the spatiotemporal dynamics of these processes. I fully agree with both points; I think this reappraisal of the photic zone concept is necessary and I found the reading of the manuscript stimulating, thought-provoking and I appreciated the historical overview.

I have only one concern with the manuscript. I found that to criticize the "photosynthesis-centricness" of the photic zone concept, the authors build a "metazoan-centric" argument, while the concerns they initially raise extend well beyond this group. For example, at L74, the authors write "This information is detected by specialised light sensitive organs or regions including eyes, ocelli, pits and dense regions of light sensitive cells." However, we know these processes take place across unicellular kingdoms as well via a large suite of photoreceptors that integrate the whole light spectrum. And this includes phototrophs for which photobiology also entails more than photosynthesis. To give one example that particularly relevant to the contexts of the review, it was recently shown in diatoms that phytochrome are crucial to sense depth in the water column and are activated to depth as low as 150 m (Duchêne et al., 2024, Nature). This is one of many examples suggesting light is used as information and possibly at intensity lower than the "1%" rule implies by a unicellular organism. I think the authors could make a better job to either state that they will focus on metazoans or try to be more inclusive of unicellular and of non-photosynthetic process in phototrophs and provide a few examples. I think choosing the latter would broaden the scope of the review and fully embrace the opportunity of bringing different research communities together in establishing a more comprehensive and precise definition of this fundamental concept.

Minor comments and suggestions

Different communities primarily use different light intensity units. In the photosynthesis community, the use of $\mu\text{mol photons m}^{-2} \text{ s}^{-1}$ is widespread. I figured that the authors considered $1 \mu\text{mol photons m}^{-2} \text{ s}^{-1} \sim 0.217 \text{ W/m}^2$, but other conversion factors are sometimes used. It would be more reader-friendly to state this equivalence somewhere early in the paper rather than only in the final paragraphs.

Table 2: The reference for "Photosynthesis minima" should be the theoretical calculations by Raven et al., 2000, Journal of the Marine Biological Association of the United Kingdom. What Hoppe et al., 2024, Nat. Comm., reports is that they observed increasing Chl a concentration in ice algae communities trigger by the return of very low light intensities, close to these theoretical calculations.

L47: "for photosynthesis to occur" should be replaced by "for the rate of photosynthesis to exceed the rate of respiration" like elsewhere.

L74: Information can be considered a resource. "Unlike photosynthesis which uses light as energy" seems more precise.

L111: I appreciate the arguments from this point forward regarding why the "1%" convention is problematic. However, providing a specific number or range for sea surface irradiance here would be useful. I personally think of maximal instantaneous irradiance at the sea surface as $\sim 1000 \mu\text{mol photons m}^{-2} \text{ s}^{-1}$. Most microalgae grow well at 1% of this intensity, further highlighting issues with this definition. While Table 2 lists a "more correct" (because nothing is exact in biology) light compensation point of $1 \mu\text{mol photons m}^{-2} \text{ s}^{-1}$, adding some context here would enhance clarity.

L164: This sentence could be improved.

Reviewer #2 (Remarks to the Author):

Major claims of the paper:

- The uppermost “euphotic zone” is conventionally considered the region where sufficient light penetrates to enable photosynthesis to occur where photic is often conflated with euphotic. The ubiquity and magnitude of photosynthesis has led to the autotroph-centric view of photic zones.
 - Photosynthesis is but one biological process that relies on natural light entering the surface ocean. The current definition of the photic zone as 1% of the surface irradiance to reach a certain depth is purely based on the optical properties of the water and ignores other equally important processes which rely on light.
 - A new definition is proposed for the photic zone as “the depth to which light from celestial light sources (sunlight, moonlight or starlight) is sufficient for photobiological processes to occur in response to these light sources”, which would be scientifically better.
 - The authors argue that a definition of the photic zone should be an absolute quantity, should vary with surface irradiance, should be biologically meaningful, and inclusive of all autotrophs and light sensitive species. It could be either representative of, or inclusive of most of taxa.
- Overall, this is a really interesting and timely paper which will be particularly relevant to marine ecologists, fisheries biologists, and ecological modellers (life-history and carbon cycling). I have just a couple of more major points which, if addressed, will make this a solid paper calling for a paradigm shift in how we think about underwater light.

Major points:

Historical continuity should be addressed: will the depth of the photic zone have to be revised as new organisms are discovered with higher sensitivity to light? How would a species-specific light niche be made comparable over time? The argument for using *Calanus* is not fully justified, especially when we know more about krill photosensitivity (see specific detail in comments below). Addressing these points will make the paper more convincing.

Minor points:

A recent work that proposed a redefinition of the twilight zone, also based on an irradiance level, should be acknowledged here: Kaartvedt, S., Langbehn, T. J., & Aksnes, D. L. (2019) Enlightening the ocean’s twilight zone. *ICES Journal of Marine Science*, 76(4), 803-812. Inclusion would enhance the argument made here, since the aims and conclusions are similar.

- Pg 1, first para., among the changes in ocean lightscapes, should include decreasing sea-ice as an important factor in local brightening
- Pg1, first para., ‘Simultaneously’ is misspelled
- Pg2, last para. Change order to improve logic to: ... daily and seasonally across latitudes...
- Pg3, second para., about primary production which is misleading. Phytoplankton demonstrate photosynthetic efficiencies only this low in oligotrophic situations. Higher values have been reported (Falkowski et al., 2017) and should be stated. Patchiness in productivity is tightly coupled with nutrient availability.
- Pg3 third para. adding weight to the sentence starting “ This presents an important...” Adding weight to this argument is that despite Arctic ocean/seas only experiencing between 1 day to 6 months a year of no direct sunlight (i.e., twilight - dysphotic) yet they are amongst the most productive waters on Earth.
- Pg3, end of last para., some mention of the role of light in the entrainment of circadian clock should be included as quite possibly a fundamental process in DVM.
- Pg3, para. 2, PAR is also taken to mean ‘Active’ rather than Available.
- Pg4, second para., expanding on ‘daylight centric’ view of the oceans with mention that most marine invertebrates are nocturnal.
- Pg4, para. 3, sentence starting: “It is also inclusive of the euphotic and dysphotic zones...” some reference to common terms here should be epi- and mesopelagic.
- Pg7, last para., sentence starting: “One approach might be to measure isolumines that are specific to the species or processes...” should cite Haefker et al 2022 *Coms Biol* as this approach was applied.
- Pg8, para., 3 first sentence should equally refer to Haefker et al 2022 (see above).
- Pg9, first para., details that *Calanus* has extreme light sensitivity. It should be mentioned that there are other species which probably contribute more to DVM (in extent and often biomass) than *Calanus* i.e., krill (Cohen et al., 2015, *Plos One*).
- Pg10, definition starting with “Measure: The depth above which light from celestial sources is sufficient”. The irradiance sensitivity in *Calanus* should be treated with caution since it is based on one study with a direct light source in a horizontal tank (Båtnes et al., 2015) - photosensitivity experiments require diffuse light sources in a vertical tank and ideally coupled to electrophysiological studies. I am not saying that the zone should not be based on *Calanus* – it’s just that our current understanding of its light induced photosensitivity should be treated with some caution. We have a better understanding of krill behaviour to light (Cohen et al., 2015 *PlosOne* and 2021 *PlosBiol*).
- Pg10, bottom sentence “...investigate the fundamental drivers of diel light niches ..” should probably cite Haefker et al 2022 *Coms Biol*.
- Table 1 be explicit that this is a list of experiments based on behaviour (I think) rather than electrophysiological.
- Table 2 statement that “...Taken as the quantity of white light required to elicit diel vertical migration reported by adult female *calanus* copepods...” is not correct. It is the amount of unidirectional light required to illicit a phototactic response,

which is not the same as DVM.

- Pg10, last paragraph, specific mention of sea ice decline as an important factor in changing underwater light fields.

Reviewer #3 (Remarks to the Author):

The manuscript by Davies and Smyth proposes a broader definition of the photic zone that is more inclusive of all biological processes influenced by light. The manuscript is well-written and clearly explains the reasoning for re-assessing conventional photic zone definitions and adopting a broader definition that integrates our changing marine lightscares. The authors propose the photic zone to encapsulate the minimum light known to elicit a photobiological response to a celestial light source, using the light sensitivity of *Calanus* sp. copepods and their diel vertical migration as a measure. Within this new definition, they provide a clearer framework to differentiate the euphotic and dysphotic zones. I thought this was an excellent contribution and broadly relevant. I do wonder about how temperature is likely interplaying with DVM of different copepod species and thus their depths and light sensitivities? However, the authors make a compelling argument and contribute an important perspective to our understanding of the dynamic nature of photic zones and the organisms that inhabit them.

RESPONSE TO REVIEWERS' COMMENTS

Reviewer #1 (Remarks to the Author):

R1: In “Redefining the Photic Zone” Davies and Smyth insightfully review the concept of the photic zone. First, they argue that historical and coincidental biases have led to a murky and biologically irrelevant definition of this ubiquitous term, based on a “1% of sea surface irradiance” rule of thumb. Second, they make the case that photobiology is not limited to photosynthesis and that other processes of enormous ecological importance (e.g., vertical migration of zooplankton) also use light. However, the current photosynthesis-centric view of the photic zone is problematic for how we define, study and discuss the spatiotemporal dynamics of these processes. I fully agree with both points; I think this reappraisal of the photic zone concept is necessary and I found the reading of the manuscript stimulating, thought-provoking and I appreciated the historical overview.

RESPONSE: We would like to thank Reviewer 1 for their positive comments.

R1: I have only one concern with the manuscript. I found that to criticize the “photosynthesis-centricness” of the photic zone concept, the authors build a “metazoan-centric” argument, while the concerns they initially raise extend well beyond this group. For example, at L74, the authors write “This information is detected by specialised light sensitive organs or regions including eyes, ocelli, pits and dense regions of light sensitive cells.” However, we know these processes take place across unicellular kingdoms as well via a large suite of photoreceptors that integrate the whole light spectrum. And this includes phototrophs for which photobiology also entails more than photosynthesis. To give one example that particularly relevant to the contexts of the review, it was recently shown in diatoms that phytochrome are crucial to sense depth in the water column and are activated to depth as low as 150 m (Duchêne et al., 2024, Nature). This is one of many examples suggesting light is used as information and possibly at intensity lower than the “1%” rule implies by a unicellular organism. I think the authors could make a better job to either state that they will focus on metazoans or try to be more inclusive of unicellular and of non-photosynthetic process in phototrophs and provide a few examples. I think choosing the latter would broaden the scope of the review and fully embrace the opportunity of bringing different research communities together in establishing a more comprehensive and precise definition of this fundamental concept.

RESPONSE: Reviewer 1 raises an important point. By focusing our arguments on metazoan photobiology we need not mean to exclude other non-photosynthesis processes in autotrophs. We have added a caveat to the end of this statement which now reads: “This information is detected by specialised light sensitive organs or regions including eyes, ocelli, pits and dense regions of light sensitive cells^{27,31-33}, and even pigments that regulate photosynthesis in autotrophs³⁴”

R1: Minor comments and suggestions

Different communities primarily use different light intensity units. In the photosynthesis community, the use of $\mu\text{mol photons m}^{-2} \text{ s}^{-1}$ is widespread. I figured that the authors considered $1 \mu\text{mol photons m}^{-2} \text{ s}^{-1} \sim 0.217 \text{ W/m}^2$, but other conversion factors are sometimes used. It would be more reader-friendly to state this equivalence somewhere early in the paper rather than only in the final paragraphs.

RESPONSE: In the first section of the review we have amended a statement to include this information: “Primary production utilises the wavelengths of light between 400 – 700 nm in the electromagnetic range known as Photosynthetically Available Radiation (PAR) or alternatively as Photosynthetically Usable Radiation (PUR)²⁰ typically measured in $\mu\text{mol photons m}^{-2} \text{ s}^{-1}$ ($1 \mu\text{mol photons m}^{-2} \text{ s}^{-1}$ is equivalent to 0.217 W m^{-2} daylight).”

R1: Table 2: The reference for “Photosynthesis minima” should be the theoretical calculations by Raven et al., 2000, Journal of the Marine Biological Association of the United Kingdom. What Hoppe et al., 2024, Nat. Comm., reports is that they observed increasing Chl a concentration in ice algae communities trigger by the return of very low light intensities, close to these theoretical calculations.

RESPONSE: Thanks for pointing this out. Edit made.

R1: L47: “for photosynthesis to occur” should be replaced by “for the rate of photosynthesis to exceed the rate of respiration” like elsewhere.

RESPONSE: Thanks for pointing this out. Edit made.

R1: L74: Information can be considered a resource. “Unlike photosynthesis which uses light as energy” seems more precise.

RESPONSE: Thanks for pointing this out. Edit made.

R1: L111: I appreciate the arguments from this point forward regarding why the “1%” convention is problematic. However, providing a specific number or range for sea surface irradiance here would be useful. I personally think of maximal instantaneous irradiance at the sea surface as $\sim 1000 \mu\text{mol photons m}^{-2} \text{ s}^{-1}$. Most microalgae grow well at 1% of this intensity, further highlighting issues with this definition. While Table 2 lists a “more correct” (because nothing is exact in biology) light compensation point of $1 \mu\text{mol photons m}^{-2} \text{ s}^{-1}$, adding some context here would enhance clarity.

RESPONSE: We have directed the reader to table 2 for context.

R1: L164: This sentence could be improved.

RESPONSE: We aren’t sure in which way the reviewer would like this sentence to be improved, however this section of the manuscript is a little aggressive towards the existing convention. We assume this is what Reviewer 1 is referring to and so have toned our language down a little.

Reviewer #2 (Remarks to the Author):

Major claims of the paper:

- The uppermost “euphotic zone” is conventionally considered the region where sufficient light penetrates to enable photosynthesis to occur where photic is often conflated with euphotic. The ubiquity and magnitude of photosynthesis has led to the autotroph-centric view of photic zones.
- Photosynthesis is but one biological process that relies on natural light entering the surface ocean. The current definition of the photic zone as 1% of the surface irradiance to reach a certain depth is purely based on the optical properties of the water and ignores other equally important processes which rely on light.
- A new definition is proposed for the photic zone as “the depth to which light from celestial light sources (sunlight, moonlight or starlight) is sufficient for photobiological processes to occur in response to these light sources”, which would be scientifically better.
- The authors argue that a definition of the photic zone should be an absolute quantity, should vary with surface irradiance, should be biologically meaningful, and inclusive of all autotrophs and light sensitive species. It could be either representative of, or inclusive of most of taxa.

Overall, this is a really interesting and timely paper which will be particularly relevant to marine ecologists, fisheries biologists, and ecological modellers (life-history and carbon cycling). I have just a couple of more major points which, if addressed, will make this a solid paper calling for a paradigm shift in how we think about underwater light.

RESPONSE: We would like to thank Reviewer 2 for their time and positive comments.

R2: Major points:

Historical continuity should be addressed: will the depth of the photic zone have to be revised as new organisms are discovered with higher sensitivity to light? How would a species-specific light niche be made comparable over time? The argument for using Calanus is not fully justified, especially when we know more about krill photosensitivity (see specific detail in comments below). Addressing these points will make the paper more convincing.

RESPONSE: We have added the following statement to the section where the Calanus threshold is introduced. “**This does not however mean that all taxa will be less sensitive to celestial light sources than Calanus. As our understanding of the sensitivity of marine taxa to light from the surface ocean deepens, more sensitive species may be discovered that bring the inclusivity of the Calanus threshold into question.**”

R1: Minor points:

A recent work that proposed a redefinition of the twilight zone, also based on an irradiance level, should be acknowledged here: Kaartvedt, S., Langbehn, T. J., & Aksnes, D. L. (2019) Enlightening the ocean's twilight zone. *ICES Journal of Marine Science*, 76(4), 803-812. Inclusion would enhance the argument made here, since the aims and conclusions are similar.

RESPONSE: Many thanks. We have included this reference when introducing the concept of using a biologically meaningful irradiance threshold as an alternative to the 1% threshold.

R2: Pg 1, first para., among the changes in ocean lightscapes, should include decreasing sea-ice as an important factor in local brightening

RESPONSE: Thanks. We have included this.

R2: Pg1, first para., 'Simultaneously' is misspelled

RESPONSE: Well spotted. Edit made. Thanks.

R2: Pg2, last para. Change order to improve logic to: ... daily and seasonally across latitudes...

RESPONSE: Thanks. Edit made.

R2: Pg3, second para., about primary production which is misleading. Phytoplankton demonstrate photosynthetic efficiencies only this low in oligotrophic situations. Higher values have been reported (Falkowski et al., 2017) and should be stated. Patchiness in productivity is tightly coupled with nutrient availability.

RESPONSE: Thanks. We have removed reference to photosynthesis efficiencies in this statement to avoid misrepresenting the facts, which are besides the point we are making. It now read "The global distribution of photosynthesis results in around 50 Gt C yr⁻¹ being drawn from the atmosphere²¹...."

R2: Pg3 third para. adding weight to the sentence starting "This presents an important..." Adding weight to this argument is that despite Arctic ocean/seas only experiencing between 1 day to 6 months a year of no direct sunlight (i.e., twilight - dysphotic) yet they are amongst the most productive waters on Earth.

RESPONSE: Good point

R2: Pg3, end of last para., some mention of the role of light in the entrainment of circadian clock should be included as quite possibly a fundamental process in DVM.

RESPONSE: Great point. We wanted to provide a range of examples of different ways in which light is used by marine organisms. We have included DVM as one example but feel a deeper exploration of the mechanistic underpinnings may detract from the main thrust of the narrative.

R2: Pg3, para. 2, PAR is also taken to mean 'Active' rather than Available.

RESPONSE: Edit made

R2: Pg4, second para., expanding on 'daylight centric' view of the oceans with mention that most marine invertebrates are nocturnal.

RESPONSE: Thanks. We have added this to the statement: "...when many marine species (especially invertebrates) are actually nocturnal"

R2: Pg4, para. 3, sentence starting: "It is also inclusive of the euphotic and dysphotic zones..." some reference to common terms here should be epi- and mesopelagic.

RESPONSE: Thanks. Edit made.

R2: Pg7, last para., sentence starting: "One approach might be to measure isolumes that are specific to the species or processes..." should cite Haefker et al 2022 Coms Biol as this approach was applied.

RESPONSE: Thanks. We have added that citation.

R2: Pg8, para., 3 first sentence should equally refer to Haefker et al 2022 (see above).

RESPONSE: Thanks. Citation added.

R2: Pg9, first para., details that Calanus has extreme light sensitivity. It should be mentioned that there are other species which probably contribute more to DVM (in extent and often biomass) than Calanus i.e., krill (Cohen et al., 2015, Plos One).

RESPONSE: Thanks. We have added the statement "While other zooplankton species may contribute more to this Diel Vertical Migration in extent and biomass, the sensitivity of Calanus to celestial light sources coupled with the high level of detail with which this sensitivity has been quantified make it ideal for measuring the depth of the photic zone"

R2: Pg10, definition starting with "Measure: The depth above which light from celestial sources is sufficient". The irradiance sensitivity in Calanus should be treated with caution since it is based on one study with a direct light source in a horizontal tank (Båtnes et al., 2015) - photosensitivity experiments require diffuse light sources in a vertical tank and ideally coupled to electrophysiological studies. I am not saying that the zone should not be based on Calanus – it's just that our current understanding of its light induced photosensitivity should be treated with some caution. We have a better understanding of krill behaviour to light (Cohen et al., 2015 PlosOne and 2021 PlosBiol).

RESPONSE: We appreciate that the experiments of Cohen et al. are somewhat more robust, however they do not report a clear indication of the minimum amount of light that the focal species of krill can respond to. This is critical for defining our threshold sensitivity. We also acknowledge that the Batnes et al paper is not perfect.

We now state in our manuscript “This does not however mean that all taxa will be less sensitive to celestial light sources than Calanus. As our understanding of the sensitivity of marine taxa to light from the surface ocean deepens, more sensitive species may be discovered that bring the inclusivity of the Calanus threshold into question. The photosensitivity of Calanus may also be refined with alternative experimental approaches.”

R2: Pg10, bottom sentence “...investigate the fundamental drivers of diel light niches ..” should probably cite Haefker et al 2022 Com Biol.

RESPONSE: Citation added.

R2: Table 1 be explicit that this is a list of experiments based on behaviour (I think) rather than electrophysiological.

RESPONSE: We have now been explicit in the legend for Table 1 “**Threshold light sensitivities for marine species behavioural responses**”

R2: Table 2 statement that “...Taken as the quantity of white light required to elicit diel vertical migration reported by adult female calanus copepods...” is not correct. It is the amount of unidirectional light required to illicit a phototactic response, which is not the same as DVM.

RESPONSE: Thanks. We have made this edit.

R2: Pg10, last paragraph, specific mention of sea ice decline as an important factor in changing underwater light fields.

RESPONSE: Thanks. We have added the statement “The open ocean is becoming darker due to increased attenuation of sunlight, moonlight, and starlight in surface waters, while coastal and offshore developments are brightening nighttime marine environments through rising artificial light emissions, and sea ice decline is brightening the polar oceans”

Reviewer #3 (Remarks to the Author):

The manuscript by Davies and Smyth proposes a broader definition of the photic zone that is more inclusive of all biological processes influenced by light. The manuscript is well-written and clearly explains the reasoning for re-assessing conventional photic zone definitions and adopting a broader definition that integrates our changing marine lightscapes. The authors propose the photic zone to encapsulate the minimum light known to elicit a photobiological response to a celestial light source, using the light sensitivity of Calanus sp. copepods and their diel vertical migration as a measure. Within this new definition, they provide a clearer framework to differentiate the euphotic and dysphotic zones. I thought this was an excellent contribution and broadly relevant. I do wonder about how temperature is likely interplaying with DVM of different copepod species and thus their depths and light sensitivities? However, the authors make a compelling argument and contribute

an important perspective to our understanding of the dynamic nature of photic zones and the organisms that inhabit them.

RESPONSE: We would like to thank Reviewer 3 for their positive comments. The Calanus threshold that underpins our proposed measure of photic zone depth is derived from light sensitivities quantified in experimental settings where water temperatures were controlled between 1 and 2°C. Reviewer three raises an important point regarding temperature effects on light sensitivity that we had not previously considered, although we do not have any empirical evidence to justify making our recommended threshold temperature adjustable.